# Clinical Picture and Risk Factors of Severe Respiratory Symptoms in COVID-19 in Children

**DOI:** 10.3390/v13122366

**Published:** 2021-11-25

**Authors:** Anna Mania, Kamil Faltin, Katarzyna Mazur-Melewska, Paweł Małecki, Katarzyna Jończyk-Potoczna, Karol Lubarski, Zuzanna Lewandowska, Agnieszka Cwalińska, Jowita Rosada-Kurasińska, Alicja Bartkowska-Śniatkowska, Magdalena Figlerowicz

**Affiliations:** 1Department of Infectious Diseases and Child Neurology, Poznan University of Medical Sciences, 60-572 Poznan, Poland; faltinkamil@interia.pl (K.F.); katarzynamelewska@ump.edu.pl (K.M.-M.); pmalecki@ump.edu.pl (P.M.); karol.lubarski@student.ump.edu.pl (K.L.); zuzanna.lewandowska@gmail.com (Z.L.); agnieszka.cwalinska@ump.edu.pl (A.C.); 2Department of Pediatric Radiology, Poznan University of Medical Sciences, 60-572 Poznan, Poland; jonczyk@ump.edu.pl; 3Department of Pediatric Anaesthesiology and Intensive Therapy, Poznan University of Medical Sciences, 60-572 Poznan, Poland; jrosada@ump.edu.pl (J.R.-K.); asniatko@ump.edu.pl (A.B.-Ś.)

**Keywords:** COVID-19, children, respiratory symptoms, comorbidities, imaging abnormalities, lactate dehydrogenase, creatinine kinase

## Abstract

Children with COVID-19 develop moderate symptoms in most cases. Thus, a proportion of children requires hospital admission. The study aimed to assess the history, clinical and laboratory parameters in children with COVID-19 concerning the severity of respiratory symptoms. The study included 332 children (median age 57 months) with COVID-19. History data, clinical findings, laboratory parameters, treatment, and outcome, were evaluated. Children were compared in the groups that varied in the severity of symptoms of respiratory tract involvement. Children who required oxygen therapy represented 8.73%, and intensive care 1.5% of the whole cohort. Comorbidities were present in 126 patients (37.95%). Factors increasing the risk of oxygen therapy included comorbidities (odds ratio (OR) = 92.39; 95% confidence interval (95% CI) = (4.19; 2036.90); *p* < 0.00001), dyspnea (OR = 45.81; 95% CI (4.05; 518.21); *p* < 0.00001), auscultation abnormalities (OR = 34.33; 95% CI (2.59; 454.64); *p* < 0.00001). Lactate dehydrogenase (LDH) > 280 IU/L and creatinine kinase > 192 IU/L were parameters with a good area under the curve (0.804-LDH) and a positive predictive value (42.9%-CK). The clinical course of COVID-19 was mild to moderate in most patients. Children with comorbidities, dyspnea, or abnormalities on auscultation are at risk of oxygen therapy. Laboratory parameters potentially useful in patients evaluated for the severe course are LDH > 200 IU/L and CK > 192 IU/L.

## 1. Introduction

According to available data, children represent 9.6–13.6% of the total number of coronavirus disease 2019 (COVID-19) cases caused by severe acute respiratory coronavirus type 2 (SARS-CoV-2) [1]. Both observational studies [2,3,4] and systematic reviews [1,5,6] report a milder course of COVID-19 in children. However, as the COVID-19 pandemic continues, the number of available reports is growing. Some describe more severe clinical course, especially in children with comorbidities, additional diagnoses, and higher inflammatory indexes [7,8]. Clinical picture of COVID-19 in children include respiratory, gastrointestinal, and also neurological symptoms [1,8]. Only 2.5% of hospital admissions and 0.8% of patients that required intensive care due to COVID-19 were children [9]. In the United Kingdom, COVID-19 fatality rate in children and adolescents was estimated at 0.2 per 100,000 in the first year of pandemics, compared to 25.5 per 100,000 for all other causes of death [10]. The more favourable course of COVID-19 in children compared to the adult population is being explained by different theories, including lower expression of SARS-CoV-2 receptor—angiotensin-converting enzyme 2 (ACE-2)—in the airways of children. Thus, the concept was recently questioned [1]. Other hypotheses state that children develop a very effective protective mechanism against SARS-CoV-2 by reprogramming innate immune cells resulting from mandatory vaccinations or frequent viral respiratory tract infections [11]. Moreover, children usually have fewer comorbidities and are less exposed to toxins than adults.

The approach to treating COVID-19 at the beginning of the pandemic was experimental, including lopinavir/ritonavir, hydroxychloroquine, and oseltamivir [3,4]. The regimens were, however, not effective in the treatment of COVID-19. As the pandemic progressed, remdesivir was introduced into the treatment along with systemic steroids, intravenous immunoglobulins and tocilizumab [4,8].

As the adult population is at least partially protected against COVID-19 by massive vaccination programmes, children under 12 years of age remain vulnerable groups for SARS-CoV-2 infection. Many countries have suffered from the increased number of SARS-CoV-2 infections in children recently. During the last week of September 2021, children represented 26.7% of the weekly reported cases in the USA. Within the last two weeks of September 2021, a 7% increase in the cumulated number of COVID-19 cases since the beginning of pandemics was reported [12].

Although the proportion of children with the severe course of COVID-19 is relatively low, the increasing number of cases in children may result in a higher number of patients with significant clinical abnormalities. Therefore, it is crucial to identify potential risk factors associated with the severe clinical course requiring more intensive management in this group of patients.

This study aims to evaluate history data, clinical and laboratory findings, and treatment of children with COVID-19 concerning the severity of respiratory symptoms.

## 2. Materials and Methods

### 2.1. Patient Enrollment

We present data from a single tertiary care centre treating children with COVID-19 from 1 March 2020 to 31 April 2021. The study included 332 children, 172 boys and 162 girls, with COVID-19 diagnosed based on clinical symptoms and laboratory confirmation of SARS-CoV-2 infection. The median age of the patient was 57 months—interquartile range (IQR) was 15–152 months. During the study period, three significant peaks of infections were observed. The first peak was noted between 1 March and 31 st August 2020. The period starting from 1 September to 31 December 2020 was considered the second peak. Beginning on 1 January 2021 to 31 April 2021, the third peak was noted.

### 2.2. Clinical Evaluation

History data, including potential SARS-CoV-2 exposure, were taken, and physical examination was performed in all patients. The decision regarding hospital admission was based on the clinical condition of the patient and possible additional diagnoses. Children with respiratory symptoms were evaluated concerning oxygen saturation (SpO_2_) < 95%. Nausea, vomiting, diarrhoea, and abdominal pain were considered gastrointestinal symptoms. Headache, seizures, and altered consciousness were regarded as neurological symptoms.

### 2.3. Laboratory Parameters

Laboratory testing involved complete blood count (CBC), C-reactive protein (CRP—the upper limit of normal—UNL < 0.5 mg/dL), procalcitonin (PCT—UNL < 0.5 ng/mL), clinical chemistry parameters, fibrinogen level, international normalised ratio (INR), D-dimer (UNL—0.5 mg/L), urinalysis, when clinically appropriate. The parameters were evaluated using standard laboratory analysers. Imaging studies were performed according to clinical indications. According to clinical findings, blood culture was performed in all patients presenting fever, the urine culture, and the pharyngeal swab. Children were also evaluated for coinfections when clinically indicated, including respiratory syncytial virus and influenza virus type A and B using real-time polymerase chain reaction tests (RT-PCR; Cobas InfluenzaA/B&RSV, Cobas; Xpert Xpress FLU/RSV, Cepheid), PCR panel for additional respiratory pathogens comprising adenovirus, rhinovirus, bocavirus, parainfluenza virus, coronavirus, Epstein-Barr virus (EBV), *Mycoplasma pneumoniae, Staphylococuss aureus, Streptococcus pneumoniae*. Stool tests for rotaviruses/adenoviruses were performed, and serological tests for parvovirus B19, EBV, *M. pneumoniae,* if clinically appropriate.

### 2.4. SARS-CoV-2 Infection Testing

SARS-CoV-2 infection was confirmed by CE IVD RT-PCR tests (various analysers) from the nasopharyngeal swab. After second-generation antigen testing for SARS-CoV-2 infection validation and approval (30 October 2020), this method was also used to confirm the diagnosis of COVID-19. The COVID-19 Ag rapid test was used (Abott). Positive tests of antigen tests were considered satisfactory, while RT-PCR was performed in the case of negative results in the presence of clinical symptoms.

### 2.5. Imaging

Chest imaging was performed according to clinical indications with lung ultrasonography as the initial test in most cases—Samsung HS40 machine (convex probe CA2-8AD: 2–8 MHz, linear probe LA3-16AD: 3–16 MHz). Chest x-ray was performed in 222 patients, while chest computed tomography in 15 patients in severe clinical condition or abnormalities on chest X-ray that required further clarification.

### 2.6. Cohort Division

The whole cohort of children with COVID-19 was divided into groups according to the severity of clinical symptoms:Mild to moderate COVID-19—symptoms of upper respiratory tract infection, mild to moderate gastrointestinal symptoms—267 children.COVID-19 related pneumonia—clinical and imaging findings of pneumonia, no need for oxygen therapy to maintain SpO_2_ > 95%—31 children.COVID-19 related pneumonia with oxygen therapy—clinical and imaging findings of pneumonia, oxygen therapy required to maintain SpO_2_ > 95%—29 children (OT).COVID-19 related pneumonia with intensive care—clinical and imaging findings of pneumonia, mechanical ventilation and intensive care required—five children (ICU).

### 2.7. Treatment

Empirical antibiotic treatment was introduced in children on admission based on clinical symptoms and the outcomes of laboratory parameters. Systemic and inhalation steroids were used in patients with severe cough or dyspnea. Remdesivir was introduced in patients over 12 years old that developed COVID-19 pneumonia with a decrease in SpO_2_. Convalescent plasma (CP) was transfused in the dose of 5–10 mL/kg in patients with moderate or severe symptoms and comorbidities, increasing the risk of potential significant abnormalities. Treatment with convalescent plasma was preceded by obtaining the legal guardian’s consent and the local Ethical Committee at the Poznan University of Medical Sciences (No. 376/20, No. 732/20 and No. 813/20).

### 2.8. Statistical Analysis

Statistical analysis was performed using Med-Calc statistical software. Continuous data were presented as the median and interquartile range (IQR) and compared using the Kruskal–Wallis test with posthoc analysis using the Conover test. The number and the percentage were given for the frequency, and the chi-square for independence was used to analyse categorical data. Differences with *p* values < 0.05 were considered statistically significant. For significant parameters, the odds ratio (OR) and 95% confidence interval (95% CI) were assessed. Logistic regression was used to further analyse the parameters with the statistically significant difference in the initial analysis and included in the multivariate analysis if the results were significant. The parameters were excluded from the multivariate analysis until only parameters with statistical significance remained. The results were presented as OR and 95% CI. Results with CI not including 1.0 were considered statistically significant. Receiver operating characteristics (ROC) were calculated for the potential of analysed parameters for the necessity of oxygen therapy. An optimal cut-off point was determined according to the highest accuracy (the lowest number of false positivity and false negativity). The area under the ROC curve (AUROC) analysis was performed for selected parameters to evaluate the prognostic value.

The approval by the Ethical Committee at the University of Medical Sciences in Poznan (No 2865/20) was obtained to perform the study.

## 3. Results

The clinical characteristic of the study groups are presented in Table 1.

Children who required oxygen therapy represented 8.73%, while patients who required admission to ICU 1.5% of the whole cohort. Comorbidities were present in 126 patients (37.95%), complex conditions affected seven children (2.11%). Coinfections were present in six children (1.81%). The majority of children were included in the group with mild to the moderate course (*n* = 267; 80.42%). No differences in age and gender were observed between the study groups. Children with mild to moderate clinical course of COVID-19 had normal BMI z-scores, while children who required oxygen therapy tended to be overweight (median z-score 2.080). Children with COVID-19 that required intensive care were rather malnourished (median z-score −2.875); *p* value = 0.0417.

Comorbidities affected a higher proportion of children with COVID-19 that required oxygen therapy (OT) or were admitted to the intensive care unit (ICU) < *p* = 0.00001. These groups contained children that had complex comorbidities.

Children from the OT and ICU groups presented fever more frequently, lasting significantly longer (*p* = 0.0138 and 0.0015, respectively) (Table 2).

Symptoms with a higher incidence in these groups include cough, dyspnea, and auscultation abnormalities (*p* values < 0.0001). However, we observed neurological signs more frequently in the OT and ICU groups (*p* < 0.00001).

No significant differences were noted between evaluated groups of children with COVID-19 in the white blood count and the number of neutrophils and lymphocytes (Table 2). Significantly higher levels of CRP, PCT was observed in the ICU group compared to the remaining groups of patients (*p* = 0.078 and 0.00015, respectively). Median D-dimers were increased considerably in OT and ICU groups (*p* = 0.0017). The level of lactate dehydrogenase (LDH) differed significantly between the groups with the highest values in the ICU patients with COVID-19 (*p* < 0.000001). The gradual increase of the median level of creatine kinase (CK), alanine aminotransferase (ALT), aspartate aminotransferase (AST), urea and creatinine between the groups. Nevertheless, the values did not exceed reference values in a clinically substantial manner.

Chest imaging showed interstitial infiltrates, ground glass opacifications or both types of lesions more frequently in children from OT and ICU groups (*p* < 0.0001). Computed tomography (CT) scans of children with severe abnormalities were presented in Figure 1A–F.

Children from the OT and ICU groups received empirical antibiotic treatment more frequently as far as the therapy is concerned. Moreover, more patients in these groups were treated with systemic and inhalation steroids and CP. Only patients from OT and ICU groups received antiviral treatment with remdesivir.

The outcome was good in the majority of children. Although the length of the hospital stay differed significantly between the groups, all children recovered and were finally discharged. Furthermore, no fatal cases were observed.

History clinical and laboratory parameters with statistically significant differences between the groups were included in the logistic regression. Multivariate analysis revealed that comorbidities, dyspnea, auscultation abnormalities, and level of LDH might be considered risk factors for oxygen therapy. The overall model significance level was *p* < 0.00001. Table 3 presents the results of multivariate logistic regression of the parameters with the significant outcome. The presence of comorbidities revealed the highest OR, as well as dyspnoea and auscultation abnormalities. LDH level was the only considerable laboratory parameter in the model.

The sensitivity and specificity of the selected cut-offs of laboratory parameters: CRP > 0.55 mg/dL, LDH > 280 IU/L, CK > 192 IU/L and D-dimers > 0.74 mg/L were presented in Table 4.

The highest sensitivity was found for LDH > 280 IU/L (88.00%), while the highest specificity was detected for CK > 192 IU/L (91.24%). Although the cut-offs for laboratory values show acceptable sensitivity and specificity, their positive predictive values (+PV) were relatively low, with the highest value for CK level > 192 IU/L: +PV = 42.9%. Figure 2 presents AUROC analysis for those parameters. The best area under the curve (AUC) was found for LDH level > 280 IU/L (0.804, *p* < 0.001). AUC for CK > 192 IU/L and CRP > 0.55 mg/dL were 0.708 and 0.728, respectively.

## 4. Discussion

Studies concerning the clinical course of COVID-19 reported a relatively mild course of the disease in children [13,14]. However, numerous reports describe significant abnormalities in a relatively small proportion of SARS-CoV-2 infected patients [15,16,17]. In our study, children who required oxygen therapy account for 8.73%, while patients who required admission to ICU 1.5% of the whole cohort of children with COVID-19.

The relation between body weight and the more severe course of COVID-19 was noted relatively early in the COVID-19 pandemic in adult patients. Moreover, it was reported that obese patients with severe symptoms of COVID-19 are younger when compared to patients with body weight within reference values. [18]. In our cohort, children from the OT group had a median BMI Z-score over 2.08, indicating a more frequent incidence of obesity. However, our ICU patients were more frequently underweight. Reports regarding low body weight in children with SARS-CoV-2 infection are scarce [19]. More studies concerning complications of COVID-19 in malnourished patients are necessary to support our findings.

Available reports describe the potential association between a severe course of COVID-19 in children and certain risk factors that include malignancies, immune deficiencies, chronic pulmonary and heart diseases, genetic and neurological disorders, diabetes and obesity [15,20,21]. Furthermore, a link between low vitamin D level and more severe course of COVID-19 was suggested [22]. Moreover, a low level of vitamin D was inversely related to high inflammatory markers. We did not evaluate vitamin D levels in our cohort. We observed a higher proportion of children with comorbidities in the OT and ICU group with a greater prevalence of neurological disorders and genetic syndromes in those groups. Complex comorbidities also affected patients from these groups. Furthermore, comorbidities were a risk factor of the need for oxygen therapy in logistic regression.

Admission to the ICU in the course of COVID-19 in children is relatively rare. Available reports state that 1.8–6.8% of children with COVID-19 require intensive treatment. In our study, five children were hospitalised in the ICU (1.5%). The most common risk factor for ICU admission is the presence of comorbidities. Thus, it is not always clear if COVID-19 or comorbidities caused the ICU admission. The necessity for ICU admission results from both comorbidities and complex medical backgrounds [16]. Other factors increasing the risk of ICU admission include male gender, signs or symptoms of lower respiratory tract infection at presentation, viral coinfection, and radiological changes suggestive of pneumonia or ARDS [1]. In our cohort, boys represented 40% of the ICU group. All children from this group presented pneumonia with characteristic radiological findings, and no coinfections were noted. Since the ICU group was small, we performed a more detailed analysis for the risk of oxygen therapy, finding the presence of dyspnea, auscultation finding and LDH level as parameters of clinical significance.

Children from the cohort were evaluated for symptoms from the respiratory tract. However, gastrointestinal symptoms, skin lesions, and neurological symptoms were also noted. We observed a significantly higher frequency of fever, cough, dyspnea, auscultation abnormalities, and neurological symptoms in children from the OT and ICU groups. Respiratory tract involvement is a typical clinical manifestation. Nevertheless, gastrointestinal tract and neurological symptoms were also described in the course of COVID-19 in children [1,23]. The latter was a more common finding in the OT and ICU groups.

Several laboratory parameters were described as potential indicators of the severe course of COVID-19 in children, comprising CRP, PCT, LDH, and D-dimers [24,25]. Children from the ICU group developed higher inflammatory indexes (CRP and PCT). Medians of biochemical parameters as ALT, AST, urea, creatinine, CK, and LDH were significantly higher in OT and ICU patients. Median D-dimers were also increasing in evaluated groups of patients. Nevertheless, only LDH level was found to be a risk factor for oxygen therapy in the logistic regression analysis. The ROC curve analysis was significant in the case of CRP, CK, LDH, and D-dimers. The cut-off of CRP only slightly exceeds the upper limit of normal. Therefore, it would be complicated to use it in clinical conditions. The remaining parameters had appropriate sensitivity and specificity; the best AUC concerned LDH > 280 IU/L. Thus, +PV was best for the level of CK > 192 IU/L. The clinical use of those parameters requires further studies. The cut-offs may seem bordeline. Nevertheless, taking into account incresed level of those parametrs in evaluationg the risk of oxygen therapy may be usefull in clinical field. However, it would probably be best to use several indexes in evaluating the risk of a more severe course of COVID-19 and monitoring the patients.

Chest imaging of patients from our cohort revealed typical abnormalities for viral infections—children with mild course affecting primarily the upper respiratory tract developed peribronchial thickening. More severe lesions were similar to those typically described in COVID-19-related pneumonia in children, including consolidations, interstitial lesions, and ground-glass opacifications. Severe lesions commonly affected the lower parts of the lungs and were usually bilateral [26]. In our study, the lesions were more frequent in children from the OT and ICU groups.

During the study period, the treatment approach has been modified according to the ongoing observations and studies. In our study, children were managed according to then-current guidelines [27]. Remdesivir was proven effective in shortening recovery in hospitalised adults with COVID-19 related respiratory tract infection [28]. The drug is recommended in hospitalised patients (12 years and older) with hypoxia in the initial phase of the disease. In our group, eight children received this type of treatment in the OT and ICU groups. It has to be stressed that children started remdesivir therapy before admission to the ICU. Attempts of the therapy with CP were described previously with varied outcome [29,30]. In our cohort, CP was administered in twenty children from all evaluated groups with the higher frequency in OT and ICU groups.

The empirical antibiotic treatment was usually implemented in children based on the clinical symptoms and the result of laboratory parameters, frequently before confirmation of SARS-CoV-2 infection was made. More severe clinical conditions with elevated inflammatory indexes were observed in the OT and ICU patients, thus increasing the frequency of empirical antibiotic treatment. Other authors described the wide use of antibiotic therapy in patients with COVID-19. Low prevalence of bacterial coinfections and excessive use of antibiotics in the COVID-19 and requires counselling to optimise patient management [31].

The sample size of our study, including children with COVID-19, was sufficient for the statistical analysis of selected clinical and laboratory parameters. It is, however, a single centre study. Therefore, the number of children admitted to the ICU was relatively low. Further studies concerning severe cases of COVID-19 are necessary to confirm the suggested risk factors that may help in the initial evaluation of patients.

## 5. Conclusions

In conclusion, the clinical course of COVID-19 was mild to moderate in most children. However, comorbidities, dyspnea, abnormalities on auscultation, and LDH level were found independent risk factors for oxygen therapy. Cut-off values of laboratory parameters in baseline evaluation of patients for potential risk of a more severe course are LDH > 280 IU/L and CK > 192 IU/L.

## Figures and Tables

**Figure 1 viruses-13-02366-f001:**
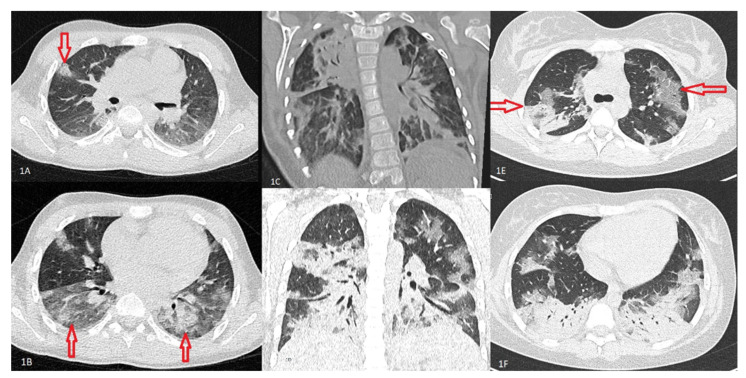
Chest computed tomography results in children with COVID-19. (**A**) Patient X—10-year old girl with Down syndrome. Chest CT scan, axial lung window, consolidation with irregular margins (arrow). (**B**) Patient X—10-year old girl with Down syndrome. Chest CT scan, axial lung window, ground-glass opacities (arrows). (**C**) Patient Y—15-year old boy with cerebral palsy. Chest CT scan, coronal lung window, large bilateral consolidations predominantly posteriorly and air bronchograms bilaterally. (**D**) Patient Z—13-year old girl, no comorbidities. Chest CT scan, coronal lung window, patchy areas of ground glass and consolidative changes. The lesions were bilateral predominantly posteriorly, reverse halo pattern in the right upper lobe. (**E**) Patient Z—13-year old girl, no comorbidities. Chest CT scan, axial lung widow, left lung diffuse ground-glass opacities with intralobular reticulation not affecting the peripheral cortex of the lung right lung ground-glass opacities and consolidative changes and air bronchograms (arrows). (**F**) Patient Z—13-year old girl, no comorbidities. Chest CT scan, axial lung widow, patchy consolidation with air bronchogram bilateral in the basal segments.

**Figure 2 viruses-13-02366-f002:**
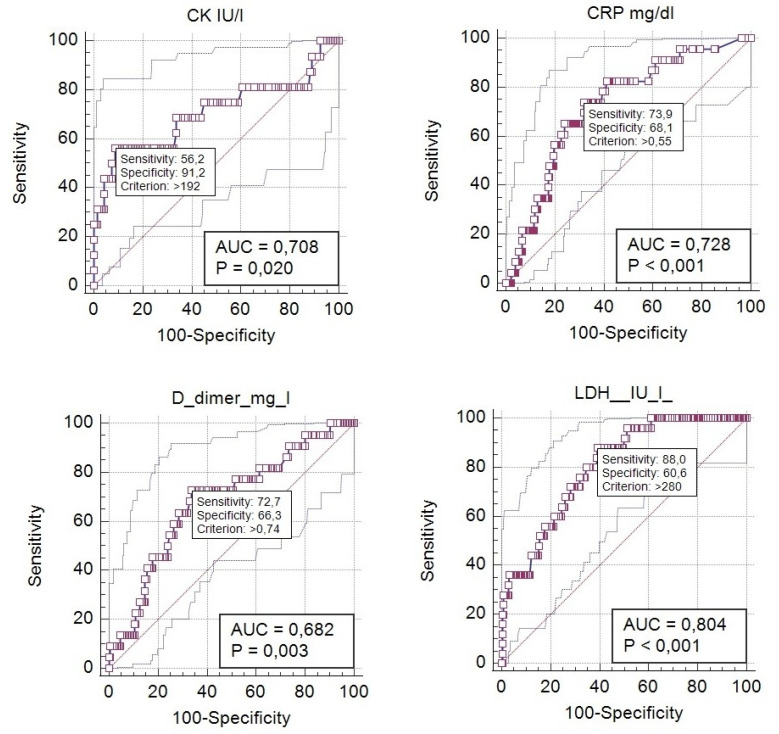
ROC curve analysis for the selected biochemical parameters and the need for oxygen therapy; AUC—area under the curve.

**Table 1 viruses-13-02366-t001:** Clinical characteristic of the study group.

Clinical Feature	COVID-19 Mild-Moderate	COVID-19 Related Pneumonia	COVID-19 Related Pneumonia with Oxygen Therapy	COVID-19 Related Pneumonia with Intensive Care	*p*-Value
Number of children (% cohort)	267 (80.42%)	31 (9.34%)	29 (8.73%)	5 (1.5%)	
Age (months) M (IQR)	62 (16–154)	30 (8–59)	84 (14–173)	21 (14–164)	0.126
Gender male/female	137/130 (51.31/48.69%)	19/12 (61.29/38.71%)	14/15 (48.28/51.72%)	2/3 (40/60%)	0.667
BMI Z-score M (IQR)	0.20 (−0.475–1.160)	0.220 (−0.130–0.460)	2.080 (1.038–2.178)	−2.875 (−5.210–0.540)	0.0417 *
Household contact (% group)	124 (46.44%)	21 (6.45%)	12 (41.38%)	1 (20.00%)	0.074
Comorbidities (% group)	86 (31.01%)	11 (32.26%)	25 (86.21%)	4 (80.00%)	<0.00001 *
Complex comorbidities	1 (0.38%)	-	3 (10.35%)	3 (60.00%)	<0.00001 *
Asthma (%)	3 (3.61%)	-	-	-	<0.0001 *
Cardiovascular diseases	8 (2.99%)	1 (3.23%)	3 (10.35%)	-	
Immune deficiency	7 (2.67%)	-	1 (3.45%)	-	
Obesity	4 (1.50%)	-	6 (20.69%)	-	
Diabetes mellitus	6 (2.25%)	1 (3.23%)	1 (3.45%)	-	
Hypertension	1 (0.38%)	1 (3.23%)	-	-	
Neurological disorders	2 (0.38%)	2 (6.45%)	3 (6.89%)	3 (60.00%)	
Genetic syndromes	-	-	5 (17.24%)	2 (20.00%)	
Other	55 (21.35%)	6 (25.80%)	8 (27.58%)	2 (20.00%)	
Coinfections (% group)	5 (1.87%)	1 (3.23%)	-	-	0.761
Pandemic peaks 1st/2nd/3rd	25/119/123	5/13/13	2/14/13	0/3/2	0.855
Length of hospital stay (days) M (IQR)	3 (2–7)	5 (4–7)	10 (6–18)	32 (29–33)	<0.000001 *

Abbreviations: M—median, IQR—interquartile range, BMI—body mass index; * *p* < 0.05.

**Table 2 viruses-13-02366-t002:** Clinical symptoms and laboratory findings in children with COVID-19 concerning the severity of symptoms.

Clinical Feature	COVID-19 Mild–Moderate	COVID-19 Related Pneumonia	COVID-19 Related Pneumonia with Oxygen Therapy	COVID-19 Related Pneumonia with Intensive Care	*p*-Value
Number of children (cohort %)	267 (80.42%)	31 (9.34%)	29 (8.73%)	5 (1.5%)	
**Clinical Signs and Symptoms—Number (group%)**					
Fever	119 (44.57%)	13 (41.93%)	21 (72.41%)	4 (80.00%)	0.0138 *
Duration of fever (days)-M (IQR)	1 (0–2)	2 (1–3)	2 (0–5)	5 (2.25–7.25)	0.0015 *
Cough	79 (29.59%)	19 (7.11%)	17 (6.37%)	4 (80.00%)	<0.0001 *
Dyspnea	17 (6.36%)	3 (9.68%)	18 (62.07%)	4 (80.00%)	<0.0001 *
Sore throat	58 (21.72%)	11 (35.48%)	5 (17.24%)	-	0.164
Gastrointestinal symptoms	83 (31.09%)	10 (32.26%)	4 (13.79%)	1 (20.00%)	0.251
Auscultation abnormalities	20 (7.49%)	3 (9.68%)	11 (37.93%)	4 (80.00%)	<0.0001 *
Skin lesions	34 (12.73%)	3 (9.68%)	5 (17.24%)	-	0.661
Neurological symptoms	6 (2.25%)	4 (12.90%)	6 (20.69%)	3 (60.00%)	<0.00001 *
**Laboratory Tests Median (IQR)**					
WBC G/l	7.86 (5.89–10.77)	9.42 (6.21–13.93)	6.25 (4.17–11.06)	8.08 (6.78–12.67)	0.128
Neutrophils G/l	3.58 (1.94–6.19)	2.89 (1.93–7.15)	2.51 (1.61–4.72)	4.42 (3.72–6.70)	0.293
Lymphocytes g/l	2.48 (1.64–4.46)	2.92 (1.62–6.22)	2.12 (0.93–3.88)	2.69 (1.92–3.69)	0.412
Platelets G/l	276 (227–360)	216 (220–355)	240 (142–353)	269 (137–253)	0.067
CRP mg/dL	0.24 (0.05–0.84)	0.46 (0.06–2.75)	0.58 (0.18–2.29)	1.41 (1.33–4.14)	0.0078 *
PCT ng/ml	0.04 (0.02–0.08)	0.05 (0.03–0.23)	0.06 (0.03–0.14)	3.82 (0.7–38.79)	0.00015 *
LDH IU/L	256 (203–310)	303 (260–354)	324 (284–406)	1627 (576–2503)	<0.000001 *
CK IU/L	91 (66–140)	125 (88–160)	157 (92–463)	216 (122–865)	0.026 *
ALT IU/L	16 (12–23)	19.5 (14–31)	32 (15–38)	45 (29–382)	0.001
AST IU/L	29 (21–39)	39 (31–49)	44 (30–52)	50 (32–452)	<0.0001 *
Urea mg/dL	21.99 (16.00–27.00)	20.00 (15.99–27.50)	21.89 (14.50–29.50)	59.00 (36.00–104.95)	0.005 *
Creatinine mg/dL	0.35 (0.25–0.52)	0.26 (0.22–0.39)	0.28 (0.18–0.48)	0.56 (0.34–2.54)	0.0078 *
Fibrinogen mg/dL	256 (218–327)	276 (214–445)	293 (240–368)	235 (123–296)	0.264
D-dimer mg/L	0.515 (0.268–0.952)	0.595 (0.380–1.070)	0.825 (0.358–1.397)	2.340 (1.715–24.350)	0.0017 *
**Chest Imaging Number (group%)**					
Chest imaging abnormalities:					
PBT II GGO II + GGO	52 (19.48%) - 1 (0.38%) -	3 (9.68%) 26 (83.87%) - -	2 (6.89%) 21 (72.41%) - 3 (10.34%)	- 4 (80.00%) - 1 (20.00%)	<0.0001*
**Treatment—Number (group%)**					
Empirical antibiotic	103 (38.58%)	30 (96.77%)	16 (55.17%)	5 (100%)	<0.0001 *
Oxygen therapy	-	-	29 (100%)	5 (100%)	<0.0001 *
Intravenous steroids	21 (7.87%)	5 (16.13%)	11 (37.93%)	5 (100%)	<0.0001 *
Inhalation steroids	8 (2.99%)	12 (38.71%)	23 (79.31%)	4 (80.00%)	<0.0001 *
Remdesivir	-	-	6 (20.69%)	3 (60.00%)	<0.0001 *
Convalescent plasma	6 (2.45%)	2 (6.45%)	8 (27.59%)	4 (80.00%)	<0.0001 *

Abbreviations: PBT—peribronchial thickening. II—interstitial infiltrates. GGO—ground-glass opacification. CRP—C-reactive protein. WBC—white blood count. ALT—alanine aminotransferase activity. AST—aspartate aminotransferase activity. CK—creatinine kinase. LDH—lactate dehydrogenase. PCT—procalcitonin. * *p* < 0.05.

**Table 3 viruses-13-02366-t003:** Logistic regression analysis for risk factors for the necessity of oxygen therapy.

Factor	Odds Ratio (OR)	95% Confidence Interval (95% CI)
Comorbidities	92.39	4.19–2036.90
Dyspnoea	45.81	4.05–518.21
Auscultation abnormalities	34.33	2.59–454.64
LDH IU/L	1.11	1.05–1.14

**Table 4 viruses-13-02366-t004:** The sensitivity and specificity of calculated cut-offs for selected biochemical parameters in the evaluation of the probability of oxygen therapy in the course of COVID-19 in children.

Parameter	Sensitivity (%)	95% CI	Specificity (%)	95% CI	+LR	-LR	+PV	-PV
CRP > 0.55 mg/dL	73.91	51.6–9.8	68.12	62.3–73.6	2.32	0.38	16.2	96.9
LDH > 280 IU/L	88.00	68.8–9.5	60.58	54.1–66.8	2.23	0.20	18.8	98.0
CK > 192 IU/L	56.25	29.9–80.2	91.24	85.2–95.4	6.42	0.48	42.9	94.7
D-dimer > 0.74 mg/L	72.73	49.8–89.3	66.32	59.1–73.0	2.16	0.41	20.0	95.5

Abbreviations: CI—confidence interval, LR—likelihood ratio, PV—predictive value.

## Data Availability

The data presented in this study are available on request from the corresponding author.

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
