# Peer review of "Clinical Picture and Risk Factors of Severe Respiratory Symptoms in COVID-19 in Children"

_viruses, 2021, doi:10.3390/v13122366_

Round 1

Reviewer 1 Report

This is a very well presented clinical paper on COVID-19 in Polish children.  There is only one caveat that I would like to note. This involved the level of LDH which might be considered a risk factor for oxygen therapy. This seems to be a borderline statistical result so an appropriate caution that this is marginally significant is in order. 

Author Response

A comment was added in the discussion regarding limitation of the laboratory parameters levels.

The text was revised for spelling mistakes.

Reviewer 2 Report

The submission by Mania et al., entitled ‘Clinical picture and risk factors of severe respiratory symptoms 2 in COVID-19 in children” is a quite informative clinical study about the risk factors of SARS-CoV-2 infection in children. The manuscript outlines the necessity and importance of the study, adopts the necessary methodology to prove the clinical conditions in children affected. However, some parts of the manuscript need revision and a few queries to be addressed for further improvement.

  1. Introduction: Should be more elaborate with examples of case studies of COVID-19 in children. Detailed information/ findings about drugs used and their effect in such cases must also be furnished.
  2. Material & methods: Should be organized with proper sub-headings.
  3. Results: Should be more descriptive.
  4. Conclusion: Should be more informative listing major findings of the study.

Other corrections,

  1. In L21-22, p value must be included
  2. L21 “95% confidence interval–95%CI” needs to be rephrased
  3. L209-211: rephrase
  4. L215: What is the average body weight? Needs to be mentioned. Parentheses in reference number mentioned should be corrected.
  5. L223: Link between Vitamin D and COVID-19 must be described from the reference, not just mentioning it.

Author Response

Thank you  very much for your comments.

The introduction was revised and suggested case studies were introduced in the description.

Materials and methods were organised in the subheadings. Results were described in detail. Conclusions were also altered according to suggestions. All the changes are marked in the text.

Other corrections;

  1. In L21-22, p was included
  2. L21 “95% confidence interval–95%CI” was rephrased
  3. L209-211: the sentence was rephrased according to suggestions.
  4. L215: The sentence was rephrased to cite the reference properly – body weight within reference values instead of average body weight
  5. L223: The link between Vitamin D and COVID-19  was described from the reference.
